# Implementation science protocol for a participatory, theory-informed implementation research programme in the context of health system strengthening in sub-Saharan Africa (ASSET-ImplementER)

Nadine Seward ![ORCID],[1] Jamie Murdoch,[2] Charlotte Hanlon ![ORCID],[3,4] Ricardo Araya,[3] Wei Gao ![ORCID],[5] Richard Harding ![ORCID],[5] Crick Lund,[3,6] Saba Hinrichs-Krapels ![ORCID],[7,8] Rosie Mayston,[8,9] Muralikrishnan Kartha,[10] Martin Prince,[8] Jane Sandall,[11] Graham Thornicroft,[1] Ruth Verhey ![ORCID],[12] Nick Sevdalis[1]

For numbered affiliations see end of article.

**Correspondence to**
Dr Nadine Seward;
nadine.seward@kcl.ac.uk

## ABSTRACT

**Objectives** ASSET (Health System Strengthening in sub-Saharan Africa) is a health system strengthening (HSS) programme involving eight work-packages (ie, a research study that addresses a specific need for HSS) that aims to develop solutions that support high-quality care. Here we present the protocol for the implementation science (IS) theme within ASSET (ASSET-ImplmentER) that aims to understand what HSS interventions work, for whom and how, and how IS methodologies can be adapted to improve the HSS interventions within resource-poor contexts.

**Settings** Publicly funded health facilities in rural and urban areas in in Ethiopia, South Africa, Sierra Leone, and Zimbabwe.

**Participants** Research staff including principal investigators, coinvestigators, field staff, PhD students, and research assistants.

**Interventions** Work-packages use a mixed-methods effectiveness–effectiveness hybrid designs. At the end of the pre-implementation phase, a workshop is held whereby the IS theme, jointly with ASSET work-packages apply IS determinant frameworks to research findings to identify factors that influence the effectiveness of delivering evidence-informed care. Determinants are used to select a set of HSS interventions for further evaluation, where work-packages also theorise selective mechanisms. *In the piloting and rolling implementation phase*, work-packages pilot the HSS interventions. An iterative process then begins involving evaluation, reflection and adaptation. Throughout this phase, IS determinant frameworks are applied to monitor and identify barriers/enablers to implementation. Selective mechanisms of action are also investigated. Implementation outcomes are evaluated using qualitative and quantitative methods. The psychometric properties of outcome measures including acceptability, appropriateness and feasibility are also evaluated. In a final workshop, work-packages come together, to reflect and explore the utility of the selected IS methods and provide suggestions for future use.

### Strengths and limitations of this study

► ASSET-ImplementER aims to advance understanding of how to use implementation science to design and evaluate health system strengthening interventions across different health systems and contexts.

► Implementation science determinant and evaluation frameworks were selected to help conceptualise what health systems strengthening interventions work for whom and how.

► Research staff working within the different work-packages are using determinant frameworks to inform their theorising of how proposed determinants interact to produce mechanisms and the identified outcomes.

► Work-packages will also validate measures of implementation outcomes in low-income and middle-income countries (LMICs), including Acceptability of Intervention Measure (AIM), Intervention Appropriateness Measure (IAM) and Feasibility of Intervention Measure.

► The research will provide recommendations for harnessing implementation science to effectively strengthen health systems in resource-poor contexts in LMICs.

Structured templates are used to organise and analyse common and heterogeneous patterns across work-packages. Qualitative data are analysed using thematic analysis and quantitative data are analysed using means and proportions.

**Conclusions** We use a novel combination of IS methods at a programmatic level to facilitate comparisons of determinants and mechanisms that influence the effectiveness of HSS interventions in achieving implementation outcomes across different contexts. The

study also contributes conceptual development and clarification at the underdeveloped interface of IS, HSS and global health.

## ETHICS AND DISSEMINATION

The ASSET-ImplementER theme is considered minimal risk as we only interview researchers involved in the different work-packages. To this effect we have received approval from King's College London Ethics Committee for research that is considered minimal risk (Reference number: MRA-20/21-21772).

## BACKGROUND

The United Nations Third Sustainable Development Goal includes a call for Universal Health Coverage (UHC), particularly within research-poor contexts within low-income and middle-income countries (LMICs).[1] Achieving UHC that explicitly addresses the availability and delivery of high-quality evidence-informed care has been identified as an urgent priority for health system strengthening (HSS) in LMICs.[2 3]

The scale of unmet need is reflected in estimates that that suggest 5 million people in LMICs who engaged with the health system died due to poor-quality health-care in 2016, and that 3.6 million deaths occurred due to people not accessing care at all.[4] Simply increasing access to care, does not necessarily improve quality of care and improve health outcomes.[5] A stark example of this issue is evident with the Janani Suraksha Yojana, a cash incentives programme targeting women who lived below the poverty line in India, to deliver in a health facility.[6 7] Despite the increased coverage of facility-based deliveries, there was no corresponding improvement with maternal and newborn outcomes. Improving population health will require not only increased access to health services, but also the provision of high-quality care.[8]

There is lack of consensus as to what constitutes health systems strengthening (HSS).[9] Initially, the term HSS was a reaction to the limitations of vertical programmes targeting specific diseases, that do not strengthen the broader health system.[9] Over time, the definition of HSS changed to one that has system-level effects and not just organisational level effects. It is now recognised that health systems are dynamic and complex sociotechnical structures, composed of multiple interacting components that are constantly adapting to changes in the local context and therefore behave unpredictably.[10] HSS to improve quality of care is about *'permanently making the system function better, and not just about filling gaps or supporting the system to produce better short-term outcomes'*.[11] HSS involves comprehensive changes to policies and regulations, organisational structures and relationships across the health system building blocks (eg, service delivery, health workforce, health information systems, access to essential medicines, financing and leadership/ governance) that motivate changes in behaviour, and/or allow more effective use of resources across multiple care platforms.[10 12]

Interventions to strengthen health systems, by their very nature, improve health outcomes by providing strategies that influence several mechanisms, both simultaneously and in isolation, at various time points and at various levels of the health system.[10] As such, HSS requires an approach to design and evaluate complex public health interventions, in real-world contexts, that accounts for the multiple interconnecting components and actors.[13] A systems-level approach to the design and evaluation of HSS interventions views a complex intervention as a system in itself, interacting with other building blocks of the underlying health system in which the intervention embeds itself, setting off reactions that may well be unexpected or unpredictable.[10] Applying this approach to the design and evaluation of HSS interventions requires an evaluation of not only their main effects, but also inputs, outputs, initial, intermediate and eventual outcomes, feedback processes and contexts within the underlying health system.[10]

### Applying implementation science concepts and methods to HSS

Implementation research is a rapidly expanding discipline that seeks to understand what, why and how interventions work in real-world settings.[14] Implementation research can consider any aspect of implementation, including understanding contextual and behavioural barriers that influence implementation efforts, the process of implementation and the evaluation of implementation efforts for outcomes such as reach, fidelity and sustainability.[14 15] By addressing implementation challenges through the participation of actors from multiple disciplines, implementation research can help to apply a systems-level approach and extract lessons that contribute to building stronger and more resilient and responsive health systems.[16]

Whereas implementation *research* seeks to understand what, why and how interventions work in real-world settings,[14] implementation science (IS) is the study of methods used to carry out this research.[17] Specifically, IS offers theories, models, frameworks and other methodologies to optimise and evaluate the implementation of evidence-informed care.[17] The methods offered through IS can inform decision-makers on how best to select and adapt HSS interventions to implement the evidence-informed care within the dynamic environments in which they work.[18] For example, IS determinant frameworks describe contextual and behavioural barriers that are known to influence implementation outcomes, and include key factors to consider in the evaluation of the process of implementation.[19] HSS interventions are selected to address specific barriers to implement the evidence-informed care.

ASSET (Health System Strengthening in Sub-Saharan Africa) is a HSS programme that aims to develop solutions that support high-quality care. The IS theme within ASSET (ASSET-ImplmentER) aims to ensure that rigorous and appropriate methodologies are applied across ASSET. Box 1 describes key IS definitions as used within the ASSET-ImplementER theme.

---

> **Box 1 Implementation science terminology used within the ASSET-ImplementER programme**
>
> Implementation strategies: methods or techniques used to enhance the adoption, implementation and sustainability of a clinical programme or practice. Other terminology includes HSS interventions, quality improvement strategies.
>
> Implementation outcomes: implementation outcomes assess how well implementation has occurred or provide insights about how this contributes to an individual's health status or other important health outcomes. Proctor *et al* articulate the following core set of eight implementation outcomes: acceptability, feasibility, uptake, penetration, cost, fidelity and sustainability.[40]
>
> Implementation effectiveness: the impact of implementation efforts on implementation outcomes.
>
> Evaluation framework: specifies implementation outcomes that can be evaluated to determine implementation effectiveness.[50]
>
> Context: any feature of the circumstances in which an intervention is conceived, developed, implemented and evaluated.[51] Contextual features are intervention specific but include (1) whole population and compositional features (varying between individuals within a population); (2) features of the physical location or geographical setting of interventions, as well as cultural, social, economic and political aspects; and (3) features affecting implementation (organisation, funding, policy, etc), as well as those directly affecting outcomes.
>
> Determinant frameworks: identify contextual barriers and/or enablers that are known to impact on the effectiveness of implementation efforts.[19]
>
> Theoretical frameworks: identify determinants of behaviours that are known to influence implementation outcomes.[38]
>
> Implementation theories/middle-range theories: describe the mechanisms behind how a proposed intervention works. These theories can also be used to identify barriers and/or enablers to change and what needs to change.[19]
>
> Programme theory: describes how a specific intervention is expected to lead to its effects and under what conditions.[52]

### Need for improved methods to design and evaluate HSS interventions in LMICs

Despite the need to account for the complexity of HSS interventions to implement evidence-informed care, current methods used to design and evaluate such interventions in LMICs typically lack robust methodologies that enable an understanding of what interventions worked, for whom and how.[20] As an example, few studies to-date have been supported by a conceptual framework (including a programme theory) explaining how an intervention is theorised to work, or a detailed process evaluation that helps to explain whether the intervention worked as intended, and if so, for whom, and under what circumstances.[20–23] There are also limitations with how studies report how the local context influences the effectiveness of HSS interventions on important implementation outcomes such as coverage, acceptability and fidelity. Importantly, much of the evidence base for HSS interventions still comes from high-income countries, with uncertain generalisability of evidence to low-resourced settings.[24]

Compounding the methodological shortcoming with HSS in LMICs, there are inconsistencies in the terminology used to describe the science of promoting and supporting the use of evidence in health and healthcare policy[25 26] and also its reporting. As an example, IS has been described variously as quality improvement, knowledge translation, knowledge transfer and even complex intervention evaluation.[26] Although there are differences between fields such IS and improvement science, they share the objective of improving the quality of healthcare delivery and therefore clinical outcomes.[17 27]

Perhaps even more important is the lack of consistency in reporting and describing the implementation strategies to improve the quality of service delivery and therefore clinical outcomes.[28 29] This results in the inability to replicate or generalise research studies to different contexts, or to allow for research synthesis such as meta-analysis or systematic reviews.[30 31] To help address this issue, the Expert Recommendations for Implementing Change (ERIC) study created a taxonomy of implementation strategies[30]—to allow researchers to apply a common language when describing how evidenced interventions are being implemented. Taxonomies for quality improvement strategies have also been created that are conceptually similar (in some cases identical) to the ERIC taxonomy.[30 32] As an example, both taxonomies include audit and feedback strategies, provider education, patient education, patient reminders and organisational change.[30 32] There is also a taxonomy for HSS interventions that includes implementation strategies, the Effective Practice and Organisation of Care (EPOC) taxonomy, that are also similar to the taxonomies for implementation and quality improvement strategies.[30 33]

There is, therefore, an urgent need to improve not only the quality of implementation research for HSS in LMICs, but also consistency in reporting the implementation strategies used to deliver evidence-informed care and improve service delivery and patient outcomes. For purposes of the ASSET-ImplementER theme, we will refer to these interventions as HSS interventions. Embedding high-quality IS methodology in HSS research, that includes consistent reporting of the HSS interventions used to deliver evidence-informed care, will help unpack the 'black box' of how such interventions work (or fail to reach their expected potential) for certain populations in a given setting.[21] The ASSET-ImplementER study aims to start addressing some of these issues.

### The National Institute of Health Research Global Research Unit on HSS in sub-Saharan Africa (ASSET) and the ASSET-implementation research theme (ASSET-ImplementER)

ASSET is a 4-year HSS research programme with a 10 month no-cost extension (2017–2022) that addresses the imperative of using implementation research with robust IS methods that use consistent reporting of the HSS interventions to deliver high-quality care. ASSET spans three healthcare platforms (primary healthcare for the integrated treatment of chronic conditions in adults,

maternal and newborn and surgical care) involving eight work-packages within four different countries within sub-Saharan Africa. A work-package can be defined as a research study that aims to address an identified need for HSS within a care platform. The overall aim of ASSET is to develop, implement and evaluate high-quality HSS interventions that are effective and sustainable. ASSET also aims to ensure care is equitable, people-centred and respectful.[34] ASSET is one of the first implementation research programmes for HSS that applies an 'effectiveness–implementation hybrid' approach, blending components of study designs that evaluate the effectiveness of interventions on patient outcomes, with study designs that focus on evaluating the effectiveness of HSS interventions on implementation outcomes.[35] This methodology allows ASSET to understand both clinical and implementation effectiveness as well as contextual factors influencing implementation outcomes, including the potential sustainability of the different HSS interventions.

The ASSET programme includes two phases of implementation research including the pre-implementation phase, and piloting and rolling implementation phase. The *pre-implementation phase* aims to understand requirements for HSS through the evaluation of the structure and function of the health system (such as arrangements for financing, governing and delivering care, and implementation strategies)[33] that may limit the potential to deliver evidence-informed and person-centred care. At the end of this phase, each work package use research findings to select a set of HSS interventions.

The *piloting and rolling implementation phase* initially pilot the interventions to identify factors that influence the implementation of the proposed interventions. After adjusting the initial programme theory and set of HSS interventions, an adaptive and iterative process is used to test the effectiveness of the set of HSS interventions on both clinical and implementation outcomes. The influence of context on the effectiveness of the HSS interventions delivering evidence-informed care is also assessed.[34] To date, ASSET has completed the pre-implementation phase of the programme and is now starting the piloting and rolling implementation phase of research. ASSET-ImplementER, the IS theme within ASSET, embeds robust methodology within and across the different work-packages to help ensure ASSET meet its overall aim and objectives.

## Aim and objectives

The overall aim of the ASSET-ImplementER theme is to advance our understanding of how to design and evaluate HSS interventions using a systems-level approach informed by IS, across different health systems and contexts. In doing so, we expect to achieve two equally important objectives for HSS in resource-poor contexts in LMICs: (1) advance our understanding of what HSS interventions work, for whom and how; and (2) improve IS methodologies to design and evaluate HSS interventions within LMIC settings.

The following specific objectives will be addressed within the ASSET-ImplementER theme:

### Pre-implementation phase

1. Contrast and compare the contextual determinants identified by each work package that influence the effectiveness of evidence-informed care.
2. Contrast and compare HSS interventions selected by each work package that have the potential to improve the effectiveness of evidence-informed care.
3. Determine how and why the investigators from the different work-packages selected the HSS interventions and implementation outcomes and compare findings between ASSET work-packages.

### Piloting and rolling implementation phase

1. Contrast and compare findings between the different work-packages on how context influences the delivery of the selected HSS interventions on standardised implementation outcome measures for acceptability, appropriateness and feasibility.
2. Evaluate the usefulness and utility of the selected IS methods/frameworks in achieving a systems-level approach in the evaluation of HSS interventions in resource-poor contexts and provide suggestions for future applications of IS to HSS.

## METHODS

### Setting

ASSET is working on three care platforms: (1) integrated primary care; (2) maternal and newborn care; and (3) surgical care, across four sub-Saharan African countries: Ethiopia, Sierra Leone, South Africa and Zimbabwe. Within the three care platforms are eight work-packages (table 1).

### Design

ASSET-ImplementER will be embedded within the timelines for ASSET (2017–2022). The work-packages use mixed-methods throughout ASSET to select HSS interventions in the pre-implementation phase and evaluate the intervention through 'effectiveness–implementation hybrid' designs in the piloting and rolling implementation phase. Hybrid designs are essential with implementation research as they blend the components of study designs used to evaluate clinical effectiveness, with those of implementation study designs that focus on the evaluation of the influence of context on the effectiveness of HSS interventions.[35]

Throughout ASSET, the ASSET-ImplementER stream uses mixed-methods including workshops, semistructured interviews and documentary analyses, to standardise, record and synthesise findings from the implementation component of the different work-packages. Findings include information from the different frameworks such as context, intervention, selected HSS interventions and implementation outcomes. Figure 1 describes the flow of methods for the ASSET-ImplementER theme.

**Table 1** Description of the ASSET work-packages for the different healthcare platforms; implementation research cuts across all of them (ASSET-ImplementER theme)

ASSET-ImplementER

| Healthcare platform | Country | Specific WP | |
|---|---|---|---|
| Primary healthcare for the integrated treatment of chronic conditions | Ethiopia | WP1. Primary care for integrated person-centred continuing care of persons with chronic non-communicable diseases including diabetes and hypertension, comorbid with common mental disorders. | |
| | South Africa | WP4. Promoting person-centred TB care. WP5. Integrated primary palliative care for persons with chronic lung disease. | |
| | Zimbabwe | WP8. Primary care for integrated treatment of persons with chronic non-communicable diseases including diabetes and hypertension, comorbid with common mental disorders. | |
| Maternal and newborn care | Ethiopia | WP2. Integrated, person-centred and high-quality maternal and newborn care across the antenatal, intrapartum, delivery and neonatal continuum. Psychosocial care for intimate partner violence is nested within this work package. | |
| | South Africa | WP6. Integrated psychosocial care/support for perinatal women experiencing depression or anxiety or exposed to domestic violence. | |
| Surgical care | Ethiopia | WP3. Increasing access and quality of surgical and dental care. | |
| | Sierra Leone | WP7. Increasing access to quality, equitable and affordable surgical care. | |

TB, Tuberculosis; WP, work-package.

To standardise methods and facilitate cross-site comparisons across ASSET, work-packages select contextual and behavioural determinants and implementation outcomes from a defined set IS frameworks. These frameworks are relevant to the programme as a whole, yet at the same time account for the specific characteristics of the different work-packages. Table 2 describes the frameworks that the various work-packages use and how this is relevant to the overall ASSET programme.

The Consolidated Framework of Implementation Research (CFIR) is a determinant framework that is used as it provides an overview of a broad range of determinants that influence implementation effectiveness, such as the inner setting (ie, characteristics of the health facility), characteristics of the intervention (eg, complexity and adaptability) and implementation processes (eg, regular feedback about progress and quality of implementation).[36] The Context and Implementation of Complex Intervention (CICI) framework is another framework that we use as it offers a detailed approach to identifying determinants from the external context (eg, sociocultural, socioeconomic, political, epidemiological, ethical and legal) that are known to influence implementation effectiveness that are particularly relevant to LMICs.[37] It is expected that both of these frameworks will provide a

detailed spectrum of determinants that are relevant to the ASSET programme. However, the frameworks are not exhaustive: any determinant identified that is not a part of either frameworks will be documented as such.

Further to contextual determinants, it is also important to understand in some detail the characteristics of the users who deliver the healthcare. Changing behaviour ingrained in both individuals working within health systems and users of the health system will help to ensure the adoption and longer-term sustainability of the HSS interventions. To address this, we will use the Theoretical Domains Framework (TDF), a determinant framework that brings together evidence-based determinants of behaviour.[38] We also use the Behaviour Change Wheel (BCW) that explicitly maps behavioural-change interventions onto determinants of behaviour identified with the TDF.[39] Each work-package selects the contextual and behavioural determinants, implementation outcomes and HSS interventions from the different frameworks that are relevant to their aims and objectives. Table 3 describes the objectives of the different IS frameworks across the different phases of ASSET. However, identifying determinants that influence the effectiveness of HSS interventions in delivering evidence-informed practices is not enough to address complexity associated with HSS.

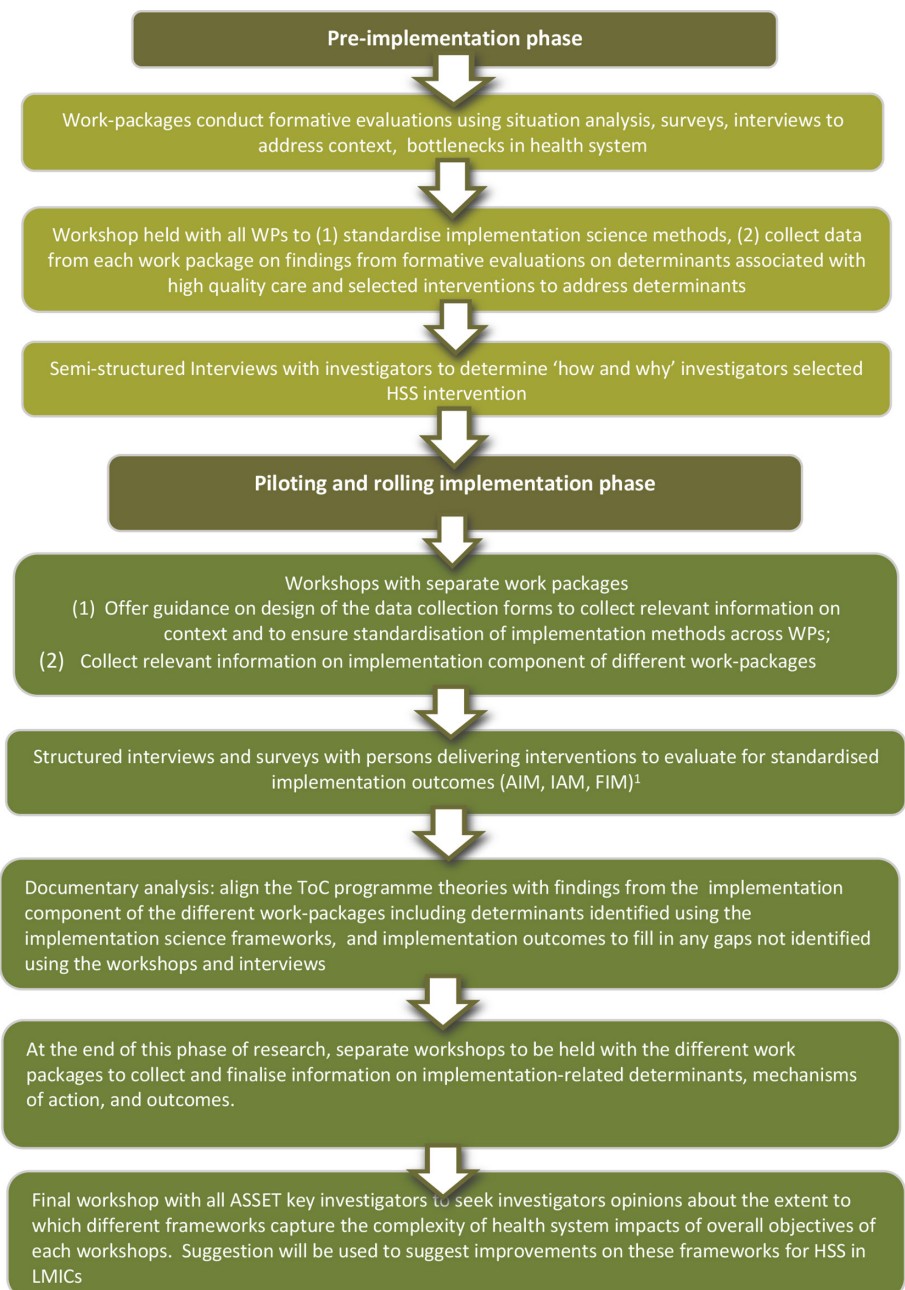

**Figure 1** Flow of methods for the ASSET-ImplementER theme. [1]*Note:* AIM, IAM and FIM have demonstrated promising psychometric properties in high-income settings.[41] AIM, Acceptability of Intervention Measure; FIM, Feasibility of Intervention Measure; HSS, health system strengthening; IAM, Intervention Appropriateness Measure; LMICs, low-income and middle-income countries; ToC, Theory-of-Change; WP, work-package.

We will therefore also explore how context influences the mechanisms introduced by the HSS interventions on implementation outcomes.

## Measures and taxonomies
### Implementation outcomes
To evaluate the effectiveness of the implementation efforts, ASSET uses a combination of implementation outcomes as defined by the framework by Proctor *et al.*[40] The selected implementation outcomes are aligned with the aims and objectives for the different work-packages.

Many measures used to evaluate implementation outcomes have not been validated, making it difficult to compare the effectiveness of alternative interventions both within and between studies.[41] However, three implementation outcomes measures including Acceptability of Intervention Measure (AIM), Intervention Appropriateness Measure (IAM) and Feasibility of Intervention Measure (FIM) have demonstrated promising psychometric properties in high-income settings.[41] Work-packages test the applicability of these measures in low-resourced settings by adapting them to their specific

**Table 2** Selected implementation science frameworks and theories used within ASSET-ImplementER

| Framework | Framework type and description | Relevance to ASSET |
|---|---|---|
| CFIR | The CFIR is a determinant framework that includes five domains (inner setting, outer setting, intervention characteristics, characteristics of individuals involved and the processes of implementation).[36] Within the five domains are 37 constructs that can behave as a barrier and/or enabler to the implementation of an intervention. | The appeal of the CFIR framework is the broad range of constructs that apply not only to individual characteristics, but also structural characteristics of the healthcare system as well as characteristics of the intervention itself.[36] |
| CICI | The CICI is both a determinant and evaluation framework that contains three dimensions (context, implementation and setting) that interact with the intervention. The contextual dimension consists of seven contextual domains (ie, geographical, epidemiological, sociocultural, socioeconomic, ethical, legal, political) with interact with one another at the micro, meso and macro levels.[37] | The appeal of this framework is the emphasis on the external contextual elements such as sociocultural, socioeconomic and political determinants. These determinants are particularly relevant in low-resource contexts. This framework also explicitly addresses complexity required with the design and evaluation of HSS interventions, through the evaluation of the interaction of context (including leadership elements), implementation and setting dimensions with the intervention dimension, at the micro, meso and macro levels. |
| TDF | The implementation of evidence-informed interventions is dependent on changing multiple behaviours of different people.[53] where the use of theory to inform behaviour change interventions has been shown to improve the implementation effectiveness.[45] We will adopt the TDF that represents a synthesis of 128 determinants of behavioural-change.[38] | Many of the barriers to HSS interventions for ASSET are associated with determinants of current and desired behaviours for example, in health worker or health service users. As an example, social norms that stigmatise common mental health conditions can act as a barrier to respectful, person-centred care. These determinants are particularly relevant to providing person-centred respectful care as well as high-quality care. |
| BCW | The implementation of evidence-informed practice is dependent on effective behaviour change interventions. The BCW is framework that includes nine intervention functions that address determinants of problematic behaviours that require changing. | To ensure longer-term sustainability, implementing evidence-informed practice for HSS will involve a degree of behavioural-change. Once determinants of the problematic or desirable behaviours have been identified using the TDF, the BCW can be applied to help select appropriate interventions that have demonstrated effectiveness. |
| Proctor *et al* Implementation outcome framework | An evaluation framework that includes the following eight implementation outcomes: acceptability, adaptability, appropriateness, feasibility, fidelity, cost, penetration and sustainability.[40] | All of these outcomes are relevant to ASSET at different time points in the implementation of the HSS interventions. As an example, the acceptability, appropriateness and feasibility of the HSS interventions are important to measure all phases of research as they can predict the longer-term sustainability of the intervention.[41] |

BCW, Behaviour Change Wheel; CFIR, Consolidated framework for Implementation Research; CICI, Context and Implementation of Complex Intervention; HSS, health system strengthening; TDF, Theoretical Domains Framework.

context. Each measure includes four questions measured on a Likert scale (completely disagree, disagree, neither agree nor disagree, agree, completely agree). Questions for each measure will be translated and back translated to ensure linguistic and cross-cultural equivalence. The questions will then be reviewed and cognitive testing with local participants performed prior to the tools being used.

## Taxonomies

Initially we use the EPOC taxonomy for HSS interventions to label the different interventions selected by work-packages.[33] If an appropriate intervention is not included in this taxonomy, work-packages use the ERIC taxonomy for implementation strategies to label the intervention.[30] To help alleviate the ambiguity in reporting implementation strategies and HSS interventions, we also report any important differences and similarities between the two taxonomies.

To provide consistency in labelling of the behavioural-change interventions, we use the taxonomy of behavioural-change interventions that is also aligned with the TDF and the BCW.[42] However, there are important limitations to applying the TDF and BCW to help select interventions

**Table 3** Application of implementation science frameworks across asset HSS interventions and research phases

| Implementation framework | Pre-implementation phase | Piloting and rolling implementation phase |
| --- | --- | --- |
| CFIR, CICI frameworks will be used to identify determinants of the implementation process including barriers/drivers and contextual influences | Identify barriers and/or enablers that may influence the delivery of high-quality care. Identified barriers are used to select a set of relevant HSS interventions. | Evaluation of selected HSS interventions to understand whether they interact with contextual barrier and/or enabler as intended. Monitor the influence of identified barriers on the effectiveness of the selected HSS interventions. Understand/explain contextual determinants that influence implementation outcomes. |
| TDF will be used to identify areas for behavioural-change interventions to enhance implementation | Identify determinants of behaviours that are known to influence healthcare professionals' ability to deliver high-quality care. Identified determinants are used to select relevant behaviour change interventions. | Evaluation of behaviour change interventions to determine whether they interact with relevant determinant of behaviour as intended. Guide the evaluation of behaviours on implementation outcomes. Understand/explain what behaviours influence implementation outcomes. |
| BCW | Used in combination with the TDF framework to help identify appropriate behavioural-change interventions. | Used in combination with the TDF to help identify any additional behavioural-change interventions. |
| Proctor *et al* framework, framework to structure the evaluation of the implementation | Identify implementation outcomes relevant to the aims and objectives of the work-package. | Evaluate relevant implementation outcomes including acceptability, appropriateness and feasibility. Ongoing assessment and adjustment of implementation outcomes at relevant stages of implementation. |

CFIR, Consolidated framework for Implementation Research; CICI, Context and Implementation of Complex Intervention; HSS, health system strengthening; TDF, Theoretical Domains Framework.

that improve the quality of people-centred care that also address issues such as illness-related stigma. As an example, these behavioural-change methods were not specifically designed to address the issues surrounding implementation research in low-resource setting such as stigma, that ASSET is focusing on. As such the behavioural-change interventions in this taxonomy may not be appropriate. To address this issue and similar issues with the other taxonomies, where the HSS interventions, quality improvement strategy or behavioural-change intervention is not described in any taxonomy, we clearly describe the intervention and highlight the issue with the selected taxonomy not describing our intervention.

### Reporting frameworks
To ensure consistent and accurate reporting of implementation studies, work-packages will apply the Standards for Reporting Implementation Studies.[43]

### Data collection procedures
#### Data collection in the pre-implementation phase
*Workshops*
At the end of the pre-implementation phase of ASSET, a 2-day workshop is held involving relevant participants from all work-packages (ie, principal investigators, coinvestigators, field staff and research assistants). The objectives of the workshop are to: (1) standardise the implementation methods being applied across the

different work-packages; and (2) collect data from each work-package on contextual and behavioural determinants associated with the delivery of high-quality evidence-informed care and the associated HSS interventions and behavioural-change interventions that were selected to help deliver this care. Findings from this exercise will help to address our first and second objectives of contrasting and comparing contextual and behavioural determinants between the different work-packages and the selected HSS interventions.

Using data collection tools that identify and describe determinants of the implementation process, participants within the different work-packages apply findings from the formative research to identify and record barriers and/or enablers to the delivery of high-quality care. Participants then use the selected determinants to theorise how potential HSS interventions, described in the EPOC taxonomy, can assist in delivering high-quality evidence-informed care relevant to their work-package. Specifically, work-packages theorise and describe the mechanisms behind how the different determinants interact to produce a consequence (ie, poor quality of care, delayed access to care, loss to follow-up, lack of people-centred care). Similar data collection methods and forms are applied to label and record determinants of behaviours that influence the delivery of evidence-informed care. The identified determinants are then used to select and record

behavioural-change interventions. Templates for the workshops can be found in online supplemental appendices 1 and 2.

### Interviews and focus groups

To understand our third objective of the pre-implementation phase of ASSET (ie, how contextual determinants influence investigators in selecting HSS interventions, implementation outcomes), the workshop is followed-up with semistructured virtual interviews involving investigators from the different work-packages. The purpose of the workshop is to help understand the 'how and why', which led them to select the HSS interventions and associated strategies to facilitate their implementation (or virtual focus groups if more feasible for country teams). The same interviews/focus groups are also used to review and finalise the findings from the workshop and to understand if there are any gaps in evidence that will require the review any additional documentation. We anticipate including a minimum of two investigators from each ASSET work-package (16 investigators minimum sample) to reach saturation of the thematic areas that emerge from their responses. A guide for the interviews can be found in online supplemental appendix 3.

### Data collection in the piloting and rolling implementation phase
### Workshops

Throughout the piloting and rolling implementation phase, workshops are held with each work-package team, involving relevant participants (principal and coinvestigators, field staff, research assistants and PhD students). The purpose of the workshops is to guide investigators on how to design the data collection tools including interview guides, using the IS determinant frameworks, HSS taxonomies and implementation outcomes. In doing so, we hope to ensure findings are standardised and therefore comparable across work-packages.

At the end of the piloting and rolling implementation phase, a separate workshop will be held with each of different work-package teams to feedback the following information collected using the data collection tools: (1) the effectiveness of HSS interventions on standardised implementation outcomes, (2) the influence of context on the effectiveness of HSS intervention in delivering evidence-informed care and (3) the direct influence of context on the mechanisms introduced by the intervention to produce change. Participants are strongly encouraged to also theorise how the identified determinants interact with one another to produce mechanisms and the identified outcomes. It is through these workshops that we will be able to standardise the labelling of contextual and behavioural determinants to allow cross site comparisons on their influence of the standardised implementation outcomes (AIM, IAM, FIM) between the different work-packages (objective 4).

At the end of the piloting and rolling implementation phase, another workshop lasting 1 day and involving investigators from all work-packages is used to reflect on the different IS methodologies and suggest improvements for further use. Specifically, we seek investigators' opinions about the extent to which different frameworks capture the complexity of how the health system impacts on the overarching problem each work-package is trying to address. Again, we anticipate including a minimum of two investigators from each ASSET work-package (16 investigators minimum sample) to reach saturation of the thematic areas that emerge from their responses. All meetings will be audio-recorded with descriptive notes of discussion. Findings from these workshops are used to suggest how frameworks can be adapted to capture this complexity (objective 5).

### Surveys

In work-packages that had available research capacity (WP1, WP2, WP5, WP6, WP8), staff delivering the interventions (ie, nurses, community health workers) are asked to complete the linguistically adapted tools for the three outcome measures, including AIM, IAM and FIM.[41] Approximately 60 staff members from each of the participating work-packages contribute to these surveys. A subset of these participants are then interviewed to discuss the usefulness of the data collection tools. In particular, we review methods to capture the influence of context on both the effectiveness of the HSS interventions in influencing implementation outcomes as well as the influence of context on influencing the mechanisms introduced by the HSS interventions on implementation outcomes. Participants are also questioned on recommendations to improve our ability to capture the influence of context on implementation outcomes (Objective 5).

### Documentary analysis

Theory-of-Change (ToC) methodology is a participatory approach involving key stakeholders that allows the articulation of the 'theory' of how a complex interventional programme will work in reality, describing the necessary interventions to bring about the change, as well as the assumptions inherent to the programme and importantly the context of implementation.[44] ASSET work-packages are developing ToCs to support their implementation and evaluation planning. ToCs are effectively programme theories, contextualised within each one of the ASSET work-packages, offering an overview of how the selected HSS interventions are theorised to achieve specific implementation outcomes. ToCs also include information on key assumptions and work-package context. Each work package, including all relevant stakeholders, develops an initial ToC in the pre-implementation phase. This programme theory is then adapted throughout the phases. We examine whether and how the ToCs align with findings from both the pre-implementation and piloting and rolling implementation phases of the different work-packages including determinants identified using the IS frameworks outlined in table 3. We will also use ToC to better articulate how context influences mechanisms

introduced by the HSS interventions described by each work-package (Objective 4).

## Data analysis

### Psychometric assessment of outcome measures AIM, IAM and FIM

The outcome measures AIM, IAM and FIM that have demonstrated promising psychometric properties in high-income countries are tested for similar properties including substantive and discriminant content validity (the extent to which a measure is judged to be reflective of a construct of interest), and interitem consistency (extent to which scale items are scored in a similar manner) by the different work-packages[45] to determine their relevance in low-resource settings. Measures are adapted to the local context, translated and back translated.

To test for substantive validity (extent to which a measure is reflective of the construct of interest) and discriminant content validity (extent to which a measure that is not supposed to be related is actually unrelated), different cadres of workers who are responsible for delivering the intervention in the different work-packages assign 31 items reflecting the three constructs, to each of the three constructs and rate their confidence in the assignments in order test. The Wilcoxon one-sample signed-rank test or t-test (as appropriate) is used to determine whether items measured the intended construct, or whether items measured a combination of constructs. Hochberg's correction is used to correct for multiple tests.[46] Intraclass correlation coefficients using a two-way mixed-effects model to assess the level of agreement in item assignments among all participants, and also within key stakeholder groups, across 31 items and for each construct.[47] The same data are also used to assess the factorial validity of the three measures, initially through exploratory factor analyses.[41] We assess interitem consistency by computing Cronbach's alpha for each of the four-item scales. For each measure, we also calculate means and SDs. Higher scores represent more favourable responses. If the measures demonstrate adequate psychometric properties, they are used to facilitate cross-site comparisons across the different work-packages.

### Data collected by ASSET-ImplementER

For each work-package, a thematic analysis is used to analyse the qualitative data collected in semi-structured interviews and workshops that identify key issues pertinent to using IS frameworks and implementation outcome measures for HSS in low-resourced contexts.

### Analysis and synthesis of data collected by different work-packages

At the end of each phase of research, data are collected from each of the work-packages (ie, contextual and behavioural determinants, mechanisms, selected HSS interventions and implementation outcomes) and entered into a template created in Excel software (vs 16.49). As an example, each work-package records information on identified barriers/enablers, associated data source, relevant IS framework, EPOC HSS interventions and implementation outcomes. Table 4 demonstrates an example of the Excel template.

We adapt the matrixed multiple case study approach to analyse and synthesise the data we record in the templates.[48] This method facilitates comparisons between relevant work-packages, organising, analysing and presenting common and heterogeneous findings across implementation sites. Such an approach also aims to create generalisable knowledge regarding what and how local factors influence implementation. The matrixed multiple case study approach uses a combination of quantitative and qualitative methods that will allow us to identify associations between specific implementation processes and contextual factors on the one hand, and implementation outcomes on the other.

Initially, data are analysed separately for each work package. Quantitative data are analysed using means and proportions. Given the heterogeneous nature of the care platforms and associated work-packages, context and selected HSS interventions, it will not be useful to compare quantitatively the influence of context on the effectiveness of HSS interventions on implementation outcomes between the work-packages. Instead, we use a qualitative approach that aims to understand why implementation outcomes were similar or different by describing the associated HSS interventions and contextual and behavioural determinants.

Results from the matrixed multiple case study approach as well as findings from the interviews and the final workshop will be triangulated to identify points of convergence

**Table 4** Sample template used to synthesise findings from the workshops and interviews

| Determinant | Data source for determinants | Implementation science framework | EPOC health system strengthening intervention | ERIC implementation strategy | Behavioural-change intervention |
|---|---|---|---|---|---|
| Illness-related stigma | Focus-group discussion with key stakeholders | CICI/TDF | Educational meetings with healthcare providers, educational materials distributed to patients and clinicians | Education and training | Motivate health workers |

CICI, Context and Implementation of Complex Intervention; EPOC, Effective Practice and Organisation of Care; ERIC, Expert Recommendations for Implementing Change; TDF, Theoretical Domains Framework.

and divergence between different work-packages. Using these methods, we analyse how each work-package defined and operationalised different IS constructs (eg, context, intervention mechanisms, HSS interventions, behaviour change techniques), and how implementation theories and frameworks were used to support specific HSS interventions throughout implementation. The final part of our analysis integrates these elements to offer an overarching understanding of how IS frameworks have been operationalised across all ASSET work-packages and whether the use of these frameworks offered added utility (from a design, implementation or evaluation perspectives).

### Patient public involvement
Patients and the public were not involved in the designing/writing protocol for this protocol. However, extensive participatory methods that involve both the patients and public will be used by the individual work-packages to design, select and evaluate the HSS interventions for ASSET.

### DISCUSSION
To our knowledge, ASSET-ImplementER is one of the first global health implementation research programmes that attempts to standardise methodologies to design and evaluate HSS interventions across different healthcare platforms and settings. In doing so, we expect to improve our understanding of what HSS interventions work for whom, by creating a compilation of HSS interventions used by the different work-packages, the associated barriers, and the effectiveness on implementation outcomes.

ASSET-ImplementER will also attempt to develop novel insights into how we can improve IS methodologies for designing implementation research on HSS in LMICs. Importantly, there is a lack of standardisation in not only the terminology used to describe implementation research and HSS, but also the methods applied to this research. By labelling contextual and behavioural determinants using established frameworks and testing psychometric properties of implementation outcome measures that have demonstrated similar properties in high-income countries, we hope to improve our ability to compare the effectiveness of different HSS interventions. Importantly, many researchers find the plethora of IS frameworks and theories overwhelming and difficult to apply in practice. Through interviews with different stakeholders, we hope to improve methods for HSS in LMICs including our ability to learn how to best apply and simplify frameworks across different health systems.

ASSET is explicitly addressing scalability through extensive participatory work, especially in the ToC workshops to develop and refine a programme theory. As an example, ToC workshops engage a wide range of stakeholders including policy makers that has a focus on ensuring scalability. Additionally, the different work-packages embed the different HSS interventions within the health systems,

using existing healthcare staff, facilities and policies. This approach is in agreement with recommendations for ensuring scalability.[49]

There are limitations to ASSET-ImplementER. Although our programme is theory-informed through the use of ToC workshops and IS determinant frameworks, we have not used IS theories or other middle-range theories to guide the design of ASSET as a programme. Indeed, many people criticise determinant frameworks (which we apply heavily within ASSET) as being 'a' theoretical. Nevertheless, given many of the work-packages teams were unfamiliar with IS methods at the time of ASSET set-up, we feel this is a pragmatic approach to applying a theory-informed approach to HSS. To mitigate the effects of this static approach to our research, we work with the different work-packages throughout the different phases of research, to theorise and conceptualise how the selected determinants interact with mechanisms introduced by the selected HSS on implementation outcomes.

The COVID-19 pandemic has impacted on the pre-implementation phase of ASSET that will have consequences in what can be achieved in the longer-term. In some cases, work-packages experienced disruptions as key research activities such as focus group meetings, interviews with patients could not take place in person. Other issues include principal investigators for the different work-packages being seconded away to COVID-19-related activities.

ASSET-ImplementER applies a system-level approach to both the design and evaluation of HSS interventions for the ASSET programme. Although there are limitations to our approach, we expect to begin advance our understanding of what HSS work for whom, and how. It is also hoped that we will start to address the issues in understanding the complexity surrounding how to effectively strengthen health systems in resource-poor contexts within LMICs.

### Author affiliations
[1]Centre for Implementation Science, Department of Health Services and Population Research, King's College London, London, UK
[2]School of Health Sciences, University of East Anglia, Norwich, UK
[3]Institute of Psychiatry, Psychology and Neuroscience, Health Service and Population Research Department, Centre for Global Mental Health, King's College London, London, UK
[4]School of Medicine, Department of Psychiatry, WHO Collaborating Centre for Mental Health Research and Capacity-Building, Addis Ababa University College of Health Sciences, Addis Ababa, Ethiopia
[5]Department of Palliative Care and Policy, King's College London, London, UK
[6]Alan J Flisher Centre for Public Mental Health, Department of Psychiatry and Mental Health, University of Cape Town, Rondebosch, Western Cape, South Africa
[7]The Policy Institute, King's College London, London, UK
[8]King's Global Health Institute, King's College London, London, UK
[9]Global Health and Social Medicine, King's College London, London, UK
[10]King's Health Economics, King's College London, London, UK
[11]Dept of Women and Children's Health, School of Life Course Sciences, FoLSM, King's College London, London, UK
[12]Research Support Centre, College of Health Sciences, University of Zimbabwe, Harare, Zimbabwe

**Contributors** NadineS wrote the first and subsequent drafts of the paper. NadineS and NickS conceptualised the idea for the paper. NadineS, NickS, JM and CH

offered insight and edited all drafts of the paper. All other authors (RA, WG, RH, CL, SH-K, RM, MK, MP, JS, AS, GT, RV) edited and offered input to various drafts of the paper.

**Funding** The study is funded by the National Institute of Health Research (NIHR) Global Health Research Unit on Health System Strengthening in Sub-Saharan Africa, King's College London (GHRU 16/136/54) using UK aid from the UK Government to support global health research. GT, WG and NS' research is also supported by the National Institute for Health Research (NIHR) Applied Research Collaboration (ARC) South London at King's College Hospital NHS Foundation Trust. GT and NS are members of King's Improvement Science, which offers co-funding to the NIHR ARC South London and is funded by King's Health Partners (Guy's and St Thomas' NHS Foundation Trust, King's College Hospital NHS Foundation Trust, King's College London and South London and Maudsley NHS Foundation Trust), Guy's and St Thomas' Charity and the Maudsley Charity. NS' research is further supported by the ASPIRES research programme (Antibiotic use across Surgical Pathways - Investigating, Redesigning and Evaluating Systems), funded by the Economic and Social Research Council. GT also receives support from the National Institute of Mental Health of the National Institutes of Health under award number R01MH100470 (Cobalt study) and by the UK Medical Research Council in relation the Emilia (MR/S001255/1) and Indigo Partnership (MR/R023697/1) awards. The views expressed are those of the authors and not necessarily those of the NHS, the NIHR, the ESRC, the MRC, the charities or the Department of Health and Social Care. CH receives support from AMARI as part of the DELTAS Africa Initiative (DEL-15-01).

**Competing interests** None declared.

**Patient consent for publication** Not required.

**Provenance and peer review** Not commissioned; externally peer reviewed.

**ORCID iDs**
Nadine Seward http://orcid.org/0000-0002-4821-9437
Charlotte Hanlon http://orcid.org/0000-0002-7937-3226
Wei Gao http://orcid.org/0000-0001-8298-3415
Richard Harding http://orcid.org/0000-0001-9653-8689
Saba Hinrichs-Krapels http://orcid.org/0000-0001-9043-8847
Ruth Verhey http://orcid.org/0000-0002-5959-1891

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
