## [Reviewer comments · BMJ Open]

ARTICLE DETAILS

TITLE (PROVISIONAL)	Implementation Science Protocol for a participatory, theory-informed implementation research programme in the context of health system strengthening in sub-Saharan Africa (ASSET-ImplementER)
AUTHORS	Seward, Nadine; Murdoch, Jamie; Hanlon, Charlotte; Araya, Ricardo; Gao, Wei; Harding, Richard; Lund, Crick; Hinrichs-Krapels, Saba; Mayston, Rosie; Kartha, Muralikrishnan; Prince, Martin; Sandall, Jane; Thornicroft, Graham; Verhey, Ruth; Sevdalis, Nick

VERSION 1 – REVIEW

REVIEWER	Kim, Bo U.S. Veterans Health Administration
REVIEW RETURNED	01-Mar-2021

GENERAL COMMENTS	Thank you for the opportunity to review this research protocol that outlines the planned study steps to (i) examine factors that influence the success of health system strengthening (HSS) interventions in sub-Saharan Africa and (ii) identify modifications to implementation science methodologies that would enhance their applicability in this context. The protocol presents the interrelated study steps, concepts guiding the research, and planned analytical approaches in a well-organized manner. Also, I found the study rationale to be convincingly articulated by the authors, specifying the context-specific knowledge- and methods-related gaps that the field needs to have filled. My specific comments below are mostly suggestions to help (i) clarify the implementation strategies and interventions being studied, and hopefully to (ii) make more consistent the descriptions of several study components throughout the presented protocol. COMMENT 1. Please consider making more clear whether HSS interventions are being regarded as strategies that are used to implement evidence-informed care practices, or as practices that are being implemented. For instance, the first item in Box 1 mentions that “HSS interventions” is an alternative terminology for “implementation strategies.” However, in some places within the protocol, HSS interventions are mentioned as entities that are being implemented (e.g., Background section, paragraph starting with “Whereas implementation research seeks to understand ...”: the third sentence states “... to design, implement, and adapt HSS interventions ...”).
---

	COMMENT 2. Please consider specifically defining what a “work package” is early in the protocol, and preferably also in the Abstract (e.g., briefly as a parenthetical statement when work packages are first mentioned in the Abstract). COMMENT 3. When discussing the different phases of implementation, please consider grouping the phases in a consistent way throughout the protocol. For example, some places in the protocol refer to piloting and implementation as belonging under one phase (e.g., Abstract), whereas other places refer to piloting being a separate phase from implementation (e.g., Table 3). COMMENT 4. Please consider providing a brief description of ASSET’s achievements to date, since starting in 2017. COMMENT 5. Please consider further specifying the extent to which the RE-AIM framework is being used in this research. For instance, in what ways is the framework guiding the research, beyond the guidance that is available from Proctor and colleagues’ implementation outcomes framework? Several related minor revision suggestions are to (i) mention the RE-AIM framework in Table 2, (ii) indicate Proctor and colleagues’ framework as having eight, not seven, implementation outcomes in Box 1, and (iii) not duplicate References 16 and 42. COMMENT 6. In the Methods section’s subsection on surveys, please consider explaining the reason for selecting the five out of eight work packages to be part of the planned survey-related study tasks. COMMENT 7. Throughout the Methods section, please consider making more explicit the study objective(s) (currently listed at the end of the Background section) that different study steps are targeting to fulfill. COMMENT 8. Although implementation efforts to help ensure sustainability are mentioned in the protocol, the study does not seem to involve explicitly assessing sustainability as an outcome. Please consider including this as a potential limitation of the study, along with a brief mention of the study team’s subsequent plans to assess sustainability (especially since sustainability is mentioned as an overall ASSET aim, in the Background section’s subsection that describes ASSET and ASSET-ImplementER). COMMENT 9. Very minor additional revision suggestions are to (i) provide the reference that is noted as “(ref)” in Table 2, (ii) specifically refer to ASSET-ImplementER within the Background section’s subsection that describes ASSET and ASSET-ImplementER, and (iii) spell out NIHR as National Institute for Health Research, instead of as a different sequence of words (e.g., within the list of abbreviations, title of the Background section’s subsection that describes ASSET and ASSET-ImplementER, when mentioning study funding).
--	---

REVIEWER	Di Ruggiero, Erica University of Toronto, Dalla Lana School of Public Health
REVIEW RETURNED	14-Mar-2021

GENERAL COMMENTS	Feedback on overall manuscript - well-written manuscript addressing an important research area in the field of implementation research as it relates to health systems strengthening. The following minor comments are proposed for your consideration and comment. Authorship: There is clear imbalance in HIC and LMIC authorship. Given the participatory nature of this work, I would have expected to see more LMIC investigators identified as co-authors. Background Consider adding reference to the relevant SDG3 in your introduction Given your focus on resource poor contexts, this section could better focus on the particular health system strengthening issues in these contexts especially since the countries included are a mix of low and middle-income countries (second paragraph) Add a line or two to introduce what ASSET is just before the text box 1 appears, and then it can be further elaborated later on in the text as the authors have done. Understandably, the authors make reference to "context" a great deal but it isn't always clear what they mean. They may wish to cite/review guidance on context and public health interventions produced by NIHR, and consider better theorizing context. It is part of one of the frameworks they are deploying, but some definition of how authors are thinking about the multidimensional nature of context would improve the article. As this is being billed as one of the first "large-scale implementation research programmes for health system strengthening", I was surprised to not see any explicit attention to scalability of interventions. Minor edits: Abstract - as written, it doesn't give the reader a clear sense of what methodology is actually proposed. Suggest that it be revised to make this clearer in the abstract. They are well-described in main part of article. second line: "programme that aims to develop and solutions that support.... - delete 'and' between develop and solutions under piloting phase - it should 'reflection' p. 12 - inconsistencies (typo)
---

VERSION 1 – AUTHOR RESPONSE

Reviewer: 1
Dr Bo Kim, U.S. Veterans Health Administration
Comments to the Author:

Thank you for the opportunity to review this research protocol that outlines the planned study steps to (i) examine factors that influence the success of health system strengthening (HSS) interventions in sub-Saharan Africa and (ii) identify modifications to implementation science methodologies that would enhance their applicability in this context. The protocol presents the interrelated study steps, concepts guiding the research, and planned analytical approaches in a well-organized manner. Also, I found the study rationale to be convincingly articulated by the authors, specifying the context-specific knowledge- and methods-related gaps that the field needs to have filled.

Thank you very much for the positive response to our research.

My specific comments below are mostly suggestions to help (i) clarify the implementation strategies and interventions being studied, and hopefully to (ii) make more consistent the descriptions of several study components throughout the presented protocol.

COMMENT 1. Please consider making more clear whether HSS interventions are being regarded as strategies that are used to implement evidence-informed care practices, or as practices that are being implemented. For instance, the first item in Box 1 mentions that “HSS interventions” is an alternative terminology for “implementation strategies.” However, in some places within the protocol, HSS interventions are mentioned as entities that are being implemented (e.g., Background section, paragraph starting with “Whereas implementation research seeks to understand ...”: the third sentence states “... to design, implement, and adapt HSS interventions ...”).

Thank you for this comment.

We have reviewed the paper to ensure the terms are being used consistently and clearly. We changed the sentence to which you referred on page 6, to the following: “The methods offered through implementation science can inform decision-makers on how best to select and adapt HSS interventions to implement the evidence-informed care within the dynamic environments in which they work.”

COMMENT 2. Please consider specifically defining what a “work package” is early in the protocol, and preferably also in the Abstract (e.g., briefly as a parenthetical statement when work packages are first mentioned in the Abstract).

Thank you for highlighting this. We have put a brief definition in the abstract (i.e. a study to address a specific need for health system strengthening in a specific context). We have also provided a definition the first time this was mentioned in the main text on page 9 “A work package can be defined as a research study that aims to address an identified need for HSS in a specified context”.

COMMENT 3. When discussing the different phases of implementation, please consider grouping the phases in a consistent way throughout the protocol. For example, some places in

the protocol refer to piloting and implementation as belonging under one phase (e.g., Abstract), whereas other places refer to piloting being a separate phase from implementation (e.g., Table 3).

Table 3 has now been revised and corrected. We have also updated a few other areas that could be seen as ambiguous in terms of the phasing of the studies (e.g. pp. 3, 10, 11, 14, 21)

COMMENT 4. Please consider providing a brief description of ASSET’s achievements to date, since starting in 2017.

We have added this sentence to page 11 “ASSET has completed the pre-implementation phase of research and is now starting the piloting and rolling implementation phase of research.”

Many of the work packages have submitted their protocols that are currently under review. Additionally, findings from the pre-implementation phase of ASSET are currently being written up.

COMMENT 5. Please consider further specifying the extent to which the RE-AIM framework is being used in this research. For instance, in what ways is the framework guiding the research, beyond the guidance that is available from Proctor and colleagues’ implementation outcomes framework? Several related minor revision suggestions are to (i) mention the RE-AIM framework in Table 2, (ii) indicate Proctor and colleagues’ framework as having eight, not seven, implementation outcomes in Box 1, and (iii) not duplicate References 16 and 42.

Thank you for highlighting this inconsistency. We have removed the RE-AIM framework from this protocol. Although we were going to use this originally, we since decided to focus on Proctor and colleagues’ framework to guide our research.

We have also updated Box 1 to state eight implementation outcomes.

We have further removed the reference for the RE-AIM framework (page 17)

We have removed the duplicate reference (number 42)

COMMENT 6. In the Methods section’s subsection on surveys, please consider explaining the reason for selecting the five out of eight work packages to be part of the planned survey-related study tasks.

Work packages were left to decide the extent to which they used implementation science methods to guide their research. This decision to include/exclude specific work packages for the section on surveys therefore was not deliberate – some work packages did not have the capacity to carry out these assessments and therefore declined to participate. This is now clearly stated on page 22.

COMMENT 7. Throughout the Methods section, please consider making more explicit the study objective(s) (currently listed at the end of the Background section) that different study steps are targeting to fulfil.

We have updated the methods to reflect the following objectives:

Pre-implementation phase

- 1. Contrast and compare the contextual determinants identified by each work package that influence the effectiveness of evidence-informed care; (now referenced on page 19)*
- 2. Contrast and compare HSS interventions selected by each work package that have the potential to improve the effectiveness of evidence-informed care; (now referenced on page 19)*
- 3. Determine how the identified contextual determinants influence ASSET investigators in selecting specific HSS interventions and associated implementation outcomes and compare findings between ASSET work-packages. (referenced on 19)*

Piloting and rolling implementation phase of ASSET HSS interventions:

- 4. Contrast and compare findings between the different work-packages on how context influences the delivery of the selected HSS interventions on standardised implementation outcomes measures for acceptability, appropriateness and feasibility; (referenced on pages 20, 21)*
- 5. Evaluate the usefulness and utility of the selected implementation science methods/frameworks in achieving a systems level approach in the evaluation of HSS interventions in resource poor contexts and provide suggestions for future applications of implementation science to HSS. (referenced on page 20)*

COMMENT 8. Although implementation efforts to help ensure sustainability are mentioned in the protocol, the study does not seem to involve explicitly assessing sustainability as an outcome. Please consider including this as a potential limitation of the study, along with a brief mention of the study team's subsequent plans to assess sustainability (especially since sustainability is mentioned as an overall ASSET aim, in the Background section's subsection that describes ASSET and ASSET-ImplementER).

This is a very fair point. We have now described this limitation in the discussion on page 27.

Due the impact of COVID-19 on ASSET we are now conducting a rolling piloting and early implementation phase only. Therefore, although we are doing everything possible to ensure sustainability, assessing for sustainability during ASSET will not be possible as this is done late in the implementation/post-implementation phase.

COMMENT 9. Very minor additional revision suggestions are to

- (i) **provide the reference that is noted as “(ref)” in Table 2,**
This has now been updated in Table 2, page 14
- (ii) **specifically refer to ASSET-ImplementER within the Background section’s subsection that describes ASSET and ASSET-ImplementER.**

We have now incorporated the ASSET-ImplementER in the abstract under the objectives, as well as on page 10.

- (iii) **spell out NIHR as National Institute for Health Research, instead of as a different sequence of words (e.g., within the list of abbreviations, title of the Background section’s subsection that describes ASSET and ASSET-ImplementER, when mentioning study funding).**

This has now been corrected on pages 2 and 9

Reviewer: 2

Dr. Erica Di Ruggiero, University of Toronto

Comments to the Author:

Feedback on overall manuscript - well-written manuscript addressing an important research area in the field of implementation research as it relates to health systems strengthening. The following minor comments are proposed for your consideration and comment.

Thank you very much for the positive reaction to our paper.

COMMENT 1. Authorship: There is clear imbalance in HIC and LMIC authorship. Given the participatory nature of this work, I would have expected to see more LMIC investigators identified as co-authors.

We agree this is an issue with this paper. Generally, within the ASSET programme there is a balance in the distribution between partners between LMICs and HICs. The leads for most of the work packages are from either South Africa, Zimbabwe, and Ethiopia. The protocol for our pre-implementation phase paper represents a better distribution of authors between LMICs and HICs (<https://www.medrxiv.org/content/10.1101/2021.01.06.20248468v1>)

The authors of this manuscript were those with expertise in implementation science, which skewed the contributions towards HIC authors in our team.

The whole ethos of ASSET is to address a clear gap in this technical expertise in LMIC investigators (in the field of global health). Indeed, the aim of this work and this protocol is to make implementation science methods more accessible to health system researchers across LMICs. To this effect, and as part of the capacity-building strategy of ASSET, we have developed and delivered training materials in implementation science, which are delivered regularly through ASSET; and we are writing up further papers, with LMIC colleagues, to further introduce the concepts and methods of this field to global health scientists and practitioners.

COMMENT 2. Background. Consider adding reference to the relevant SDG3 in your introduction. Given your focus on resource poor contexts, this section could better focus on the particular health system strengthening issues in these contexts especially since the countries included are a mix of low and middle-income countries (second paragraph)

Thank you for highlighting this. We have updated the first two paragraphs to introduce the 3rd sustainable development goal and discuss these contexts (pg. 5).

The United Nations Third Sustainable Development Goal (SDG3) includes a call for Universal Health Coverage (UHC), particularly within research poor contexts within LMICs.(1) Achieving UHC that explicitly addresses the availability and delivery of high-quality evidence-informed care has been identified as an urgent priority for health system strengthening in low- and middle-income countries (LMICs).(2, 3)

The scale of unmet need is reflected in estimates that suggest 5 million people in LMICs who engaged with the health system died due to poor-quality health care in 2016, and that 3.6 million deaths occurred due to people not accessing care at all.(4) Simply increasing access to care, does not necessarily improve quality of care and improve health outcomes.(5) A stark example of this issue is evident with the Janani Suraksha Yojana (JSY), a cash incentives programme targeting women who lived below the poverty line in India, to deliver in a health facility.(6, 7) Despite the increased coverage of facility-based deliveries, there was no corresponding improvement with maternal and newborn outcomes. Improving population health will require not only increased access to health services, but also the provision of high-quality care.(8)

COMMENT 3. Add a line or two to introduce what ASSET is just before the text box 1 appears, and then it can be further elaborated later on in the text as the authors have done.

We have added a few sentences to introduce ASSET and ASSET-ImplementER.” ASSET (Health System Strengthening in Sub-Saharan Africa) is a health system strengthening (HSS) programme that aims to develop and solutions that support high-quality care. The implementation science (IS) theme within ASSET (ASSET-ImplementER) aims to ensure rigorous and appropriate methodologies are applied across ASSET”

COMMENT 4. Understandably, the authors make reference to "context" a great deal but it isn't always clear what they mean. They may wish to cite/review guidance on context and public health interventions produced by NIHR, and consider better theorizing context. It is part of one of the frameworks they are deploying, but some definition of how authors are thinking about the multidimensional nature of context would improve the article.

Thank you for this comment. Context is key concept within health system strengthening. We have described our approach to addressing context in more detail on page 20 “Participants then use the selected determinants to theorise how potential HSS interventions described in the EPOC taxonomy, can assist in delivering high-quality evidence-informed care relevant to their work package. Specifically, participants describe the mechanisms behind how the different determinants interact to produce a consequence (i.e. poor quality of care, delayed access to care, loss to follow-up, lack of people centred care). Similar data collection methods and forms are applied to label and record determinants of behaviours that influence the delivery of evidence-informed care.

On page 21 we also state that “Participants are strongly encouraged to also theorise how the identified determinants interact with one another to produce mechanisms and the identified outcomes.”

We further describe in the conclusions on page 27 some of the issues with determinant frameworks being ‘atheoretical and discuss how we try to mitigate this “There are limitations to ASSET-ImplementER. Although our programme is theory-informed through the use of ToC workshops and implementation science determinant frameworks, we have not used implementation science theories or other middle-range theories to guide the design of ASSET as a programme. Indeed, many people criticise determinant frameworks (which we apply heavily within ASSET) as being atheoretical. Nevertheless, given that many of the work-packages teams were unfamiliar with implementation science methods at the time of ASSET set-up, we feel this is a pragmatic approach to applying a theory-informed approach to HSS. To mitigate the effects of this static approach to our research, we work with the different work-packages throughout the different phases of research, to theorise and conceptualise how the selected determinants interact with mechanisms introduced by the selected HSS on implementation outcomes.”

We have also described the limitation of context on page 16, and how we will explore “how context influences the mechanism introduced by the HSS interventions on implementation outcomes”

COMMENT 4. As this is being billed as one of the first "large-scale implementation research programmes for health system strengthening", I was surprised to not see any explicit attention to scalability of interventions.

Our approach fits with recommended steps to ensuring scalability. In particular, the different work packages involve extensive participatory work, especially the ToC, which is done with stakeholders, including policy makers, which has a focus on ensuring scalability. Additionally, the different work packages use existing healthcare staff, stationary, facilities, and policies as part of the health system strengthening interventions with the intention that these will be brought to scale.

We have added a short paragraph in the discussion to this effect on page 27.

COMMENT 5. Minor edits:

Abstract - as written, it doesn't give the reader a clear sense of what methodology is actually

proposed. Suggest that it be revised to make this clearer in the abstract. They are well-described in main part of article.

The word limit for the abstract influenced what we could describe for such a complex programme. We have modified this on pages 3-4, but it is now over the word limit.

second line: "programme that aims to develop and solutions that support.... - delete 'and' between develop and solutions

Thank you for noticing this! This has now been updated.

under piloting phase - it should 'reflection'

Thank you for noticing this – now updated

p. 12 - inconsistencies (typo)

This has been corrected

VERSION 2 – REVIEW

REVIEWER	Kim, Bo U.S. Veterans Health Administration
REVIEW RETURNED	06-Jun-2021

GENERAL COMMENTS	Thank you for the opportunity to review this revised version of the protocol. I found this revision to be very responsive to reviewers' comments on the protocol's initially submitted version. I have no further comments to offer to the authors.
---

REVIEWER	Di Ruggiero, Erica University of Toronto, Dalla Lana School of Public Health
REVIEW RETURNED	17-Jun-2021

GENERAL COMMENTS	Thank you for attending the comments so thoroughly. I have no further suggestions.
--